# Artificial accommodating intraocular lens powered by an ion polymer-metal composite actuator

Tetsuya Horiuchi[1]*, Toshifumi Mihashi[2,3], Sujin Hoshi[2], Fumiki Okamoto[2], Tetsuro Oshika[2]

1 Nanomaterials Research Institute, National Institute of Advanced Industrial Science and Technology (AIST), Ikeda-city, Osaka, Japan, 2 Department of Ophthalmology, Faculty of Medicine, University of Tsukuba, Tsukuba-city, Ibaraki, Japan, 3 Department of Orthoptics, Faculty of Medical Technology, Teikyo University, Itabashi, Tokyo, Japan

* tetsu-horiuchi@aist.go.jp

**Data Availability Statement:** All data generated or analyzed during this study are included in this article.

**Funding:** This work was supported by JSPS KAKENHI Grant-in-Aid for Scientific Research in

## Abstract

The current method of controlling the focus of an accommodating intraocular lens is based on ciliary muscle contraction and cannot be used in older patients with presbyopia. We aimed to develop a dynamically accommodating intraocular lens powered by a membrane-shaped ion polymer metal composite actuator that is thin enough to be inserted in the eye. This study addresses two key problems identified in our previous accommodating intraocular lens prototype: the lack of repeatability due to the use of swine lenses instead of artificial lenses and the occurrence of a sixth order aberration. Thus, we present a new accommodating intraocular lens design and a method to transfer energy to actuators. To accommodate lens deformation and depth of focus, we used a membrane-shaped ion polymer metal composite actuator, thin enough to be inserted in the eye, and used an artificial silicone lens. To prevent the sixth order aberration, we included a ring between the ion polymer metal composite actuator and the lens. Different voltage patterns were applied to the IPMC actuator and changes in focus were observed. We were able to obtain repeatability and prevent the sixth order aberration. The dioptric power changed to ±0.23 D when ±1.5 V was used; however, at >1.5 V, a large accommodating range occurred, in addition to astigmatic vision. Thus, we have developed a novel prototype that is completely artificial, allowing reproducible and repeatable results. Visual accommodative demands were successfully met; however, although astigmatic vision was lessened, it was not completely eradicated.

## Introduction

Cataracts are one of the most common visual disorders which are responsible for 33% of all cases of blindness [1]. A cataract cannot be managed medically, making surgery necessary. The white-clouded cataract lens is surgically removed and replaced with an IOL.

There are three types of IOLs; monofocal, multifocal, and accommodating IOL. The specifications of each are given in Table 1.

Innovative Areas "Science of Soft Robot" project (grant numbers JP18H05470 and 18K18411) and T-CReDO, Tsukuba Clinical Research & Development Organization (grant number A17-83).

**Competing interests:** The authors have declared that no competing interests exist.

A monofocal IOL is the most commonly used IOL. It is a fixed lens, with one focal length. However, the patient's vision is limited to the range of the depth of focus [2]. The subjective depth of focus has been reported to be 0.59 diopters (D) for a pupil diameter of 4 mm. Due to this limitation, patients are required to wear convex glasses to perform activities of daily living or work involving near vision.

Hoffer et al. [3] developed a multifocal IOL that addresses issues with monofocal IOL. The surface of this IOL contains a combination of two or three different focal lenses, hence the patient will be able to see far and near objects. However, there are two major concerns with a multifocal lens; aberration and halo with glare phenomenon. Normally, a lens has a spherical surface of one radian. The surface of a multifocal IOL has a spherical surface of multiple radians. Consequently, patients with multifocal IOLs have to visualize not only with focused sight but also non-focused sight, reducing the contrast of sight. Halo and glare phenomena occur when the patient sees light under dark conditions which reduces contrast and scatters the light. These occur because the edges between the near-focus lens and far-focus lens are far from the ideal lens shape.

These challenges can be difficult to address due to the fundamental structure of a multifocal IOL. Some researchers suggest the use of monofocal IOLs that can control the focal length similar to human eyes. This type of IOL is called an accommodating IOL. Three general types of accommodating IOLs are shown in Fig 1. [4].

Crystalens and Trulign (Bausch & Lomb Inc. Bridgewater, NJ, USA) are widely used moving lens type of IOLs. These IOLs have springs on each side that are attached to the side walls of the capsula lentis. When the ciliary muscle contracts, the capsula lentis is deformed and the spring is bent, hence the IOL position changes. Consequently, the focus of the patient's eye changes. Akkolens International BV (Breda, The Netherlands) developed a sliding-type IOL

**Table 1. Features of IOLs.**

|  | Monofocal IOL | Multifocal IOL | Accommodating IOL |
|---|---|---|---|
| Focus points | 1 | Mainly, 2 or 3 | 1 |
| Convex glasses | Needed | Not needed | Not needed |
| Remarks column | - | Halo and glare phenomenon | - |

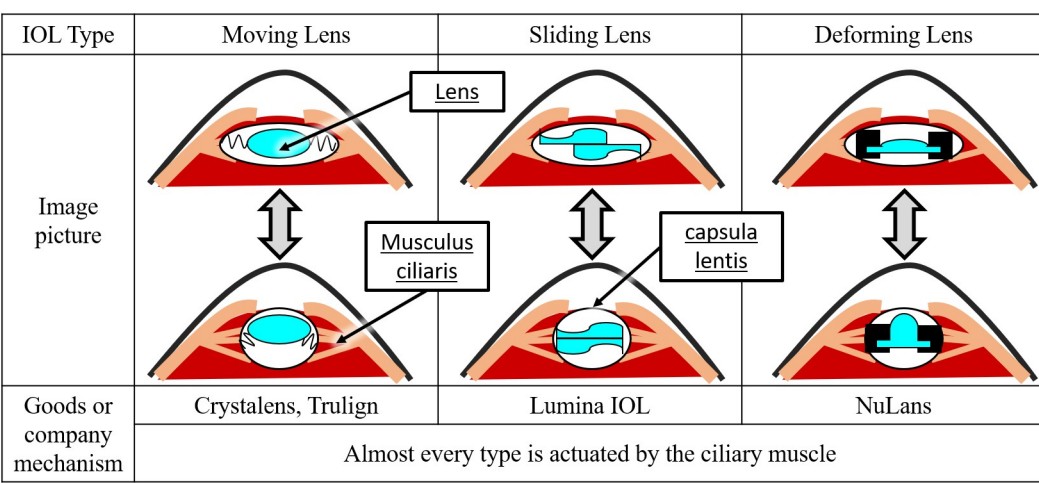

**Fig 1. Three general series of accommodating intraocular lenses (IOLs) [4].**

called Lumina IOL [5]. This IOL is also attached to the capsula lentis and contains two lenses that slide with the deformation of the capsula lentis in the ciliary muscle. Fig 1 shows the shapes of these lenses along with their focal change mechanisms.

NuLens Ltd. proposed to change the focus by deforming a soft lens [6]. This is also set in the capsula lentis. When the ciliary muscle contracts, the surface of the soft lens becomes deformed and the focal length changes. The Flex Optic IOL (Advanced Medical Optics, Inc., Irvine, CA, USA) is a single-optic IOL [5] with a flexible lens and an arch. When the ciliary muscle contracts, the arch is deformed and changes the shape of the lens surface. McCafferty et al. [7] suggested a similar deforming-type IOL that uses an arch and a silicone gel. The FluidVision Lens (Power Vision Inc., Belmont, CA, USA) drives fluid with a polymer-matched refractive index [5]. This lens also controls the focal length by changing the lens surface with the ciliary muscle.

The conventional accommodating IOL can be used to control the focus of the lens with the patient's ciliary muscle; however, a significant problem exists. The accommodating ranges vary widely from patient to patient. This phenomenon cannot be resolved by these methods while the patients' ciliary muscles are used for accommodating.

Some researchers have already proposed a new accommodating IOL. Peng et al. [8] suggested an accommodating IOL that uses a conductive liquid moved by an electric field. The surface of the liquid is changed by the electric field, which changes the focus. Hasan et al. [9] suggested an accommodating IOL that uses a shape memory alloy spring controlled by electric heat. Vdovin et al. [10] suggested an accommodating IOL that uses a liquid crystal lens deformed by conductive metal that is moved by an electrical field. Wei et al. [11] used dielectric elastomers to control the focus of the lens. These methods can be used for patients with presbyopia. However, a major drawback with these methods is electrical leakage. Electrolysis of water occurs above 1.5 V. To ensure safety, IOL systems should not be used over 1.5 V. However, the above method requires a higher voltage than 1.5 V.

## Previous work

Our aim was to develop a new control system for changing the focus of an accommodating IOL with a polymer actuator. The concept of a lens control system with a polymer actuator is not novel. Carpi et al. [12] and Maffi et al. [13] suggested a tunable silicone lens, which is driven by dielectric elastomer actuators. Moreover, Kikuchi et al. [14] suggested a variable-focal length lens, which is controlled by an ion polymer metal composite (IPMC) actuator. However, their aim was to create a lens for camera-like devices, not for accommodating an IOL.

Our team is studying accommodating IOLs, and our previous accommodating system is shown in Fig 2.

To control the changing focus, we used an IPMC actuator that can induce accommodative changes in a targeted lens. In the previous study, we used swine crystalline lenses. However, it was difficult to insert the lens into the eye due to its shape and the size of the IPMC actuator which is over 15 mm.

## Materials and methods

### Future model of the accommodating ion polymer metal composite

**Design.** We considered two types of accommodating IOLs: deforming lens type and moving lens type (both are shown in Fig 1).

The substantial merit of the moving lens is its strength for high-order aberration since the shape of the lens is fixed and the movement only occurs while positioning the lens. However, it is hard to get enough accommodating range.

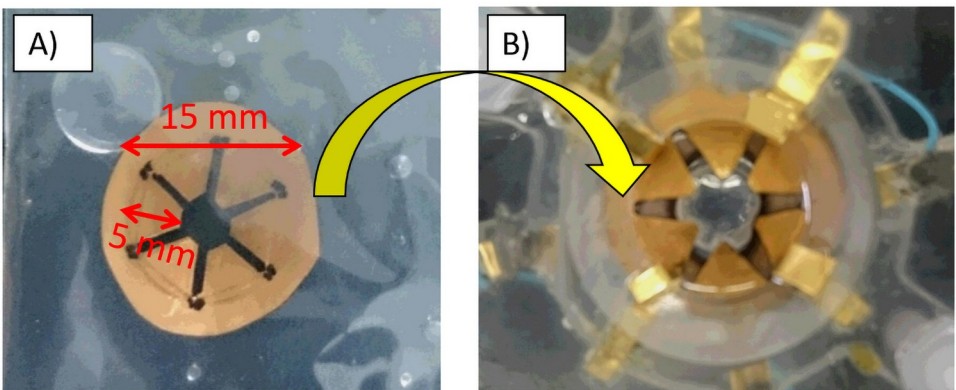

**Fig 2. Our previous accommodating system with ion polymer metal composite (IPMC) actuator.** A) IPMC actuator. B) IPMC actuator with a swine lens.

We adopted the deforming lens type for our purposes. This model consists of the lens, the IPMC actuator, cover, and the ring (Fig 3). The lens and the ring are placed between the cover and the IPMC actuator which are then fixed by sutures. The moving parts of the IPMC actuator do not directly touch the lens. The ring touches the lens to prevent high order aberrations. In the previous research, we had sixth order aberrations (Fig 4) as there were six moving parts in the IPMC actuator. Thus, we opted to utilize the ring to prevent aberration.

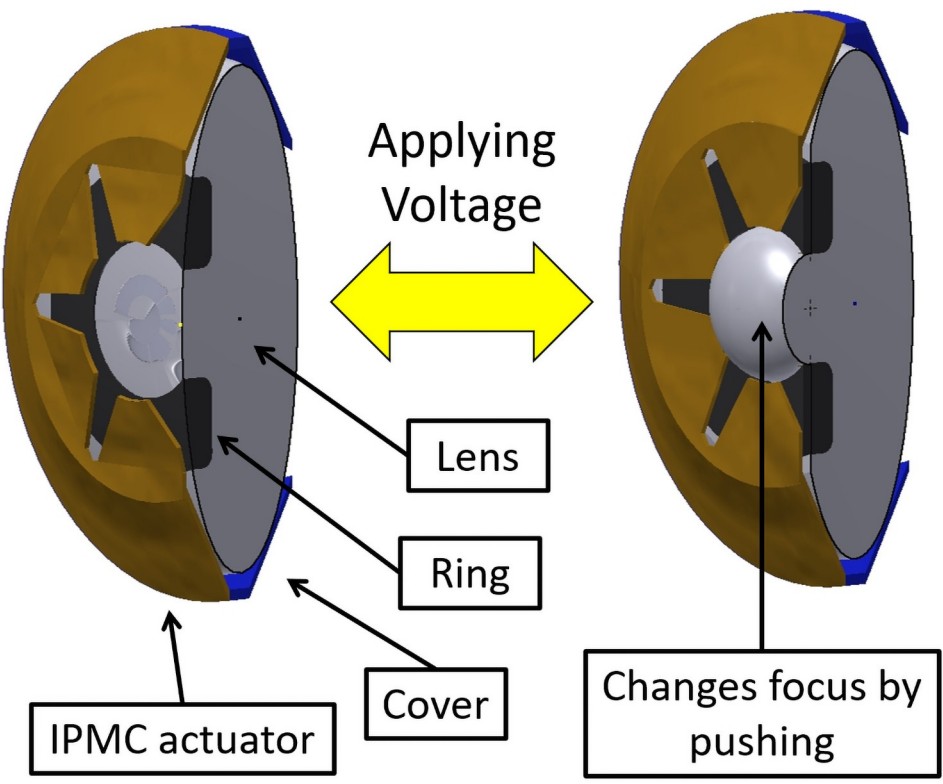

**Fig 3. Suggested accommodating system.**

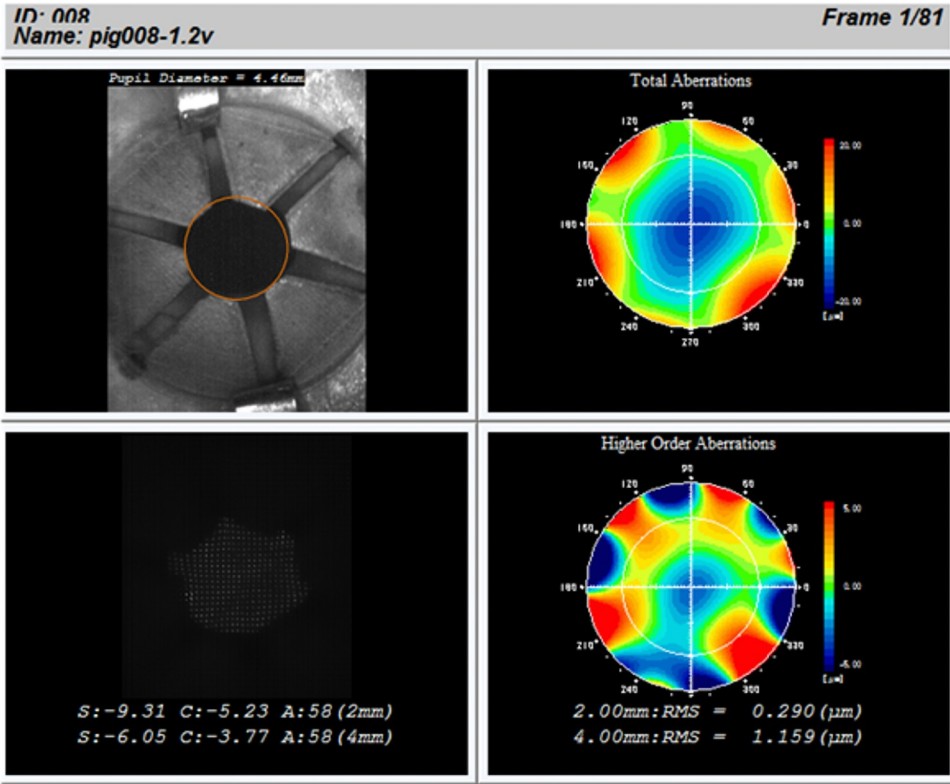

**Fig 4. Higher order aberrations from results of previous experiment.**

**Power supply.** There are two concerns with an accommodating IOL moved by electric power; first, energy transfer to the actuator, and second, control of the accommodating IOL by the patient. To address these concerns, we suggest transference from the eye socket.

We proposed the system shown in Fig 5 that used the convergence of the eye.

A part of the IPMC actuator protruded outside the augen from the sclera. An electrical tool, like an electrode or capacitor, was placed on the surface of the IPMC actuator labeled as inner connector in Fig 5. In addition, another electrical tool was placed on the surface of the eye socket. Usually, these two tools are not connected and do not transfer electrical energy. However, once the patient focuses on close objects the eyes are set close and, being attached to both electrical tools, electrical energy is transferred. A simple logic circuit can be used to avoid this issue. For example, when a patient wants to focus to the right or left, the electrical tool is attached to one side of the eye; however, the tool on the other side is not attached. Without a

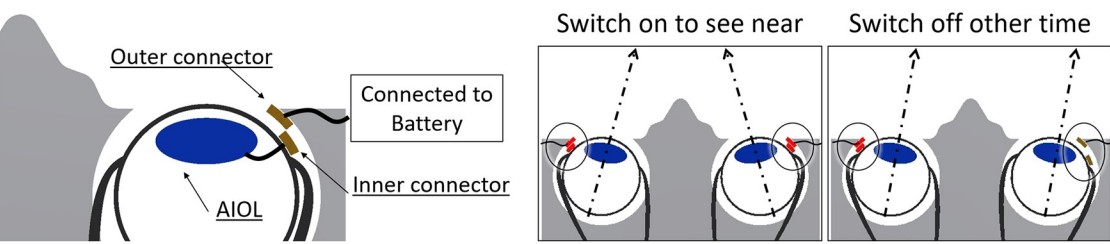

**Fig 5. Energy supply and control system.**

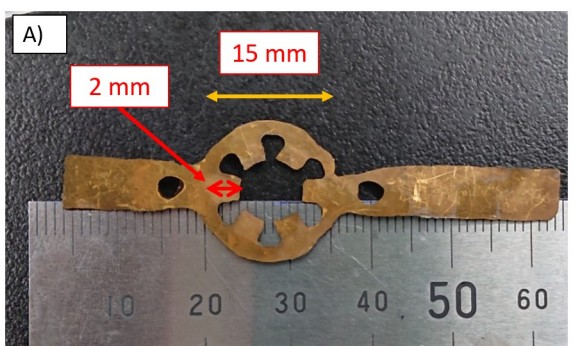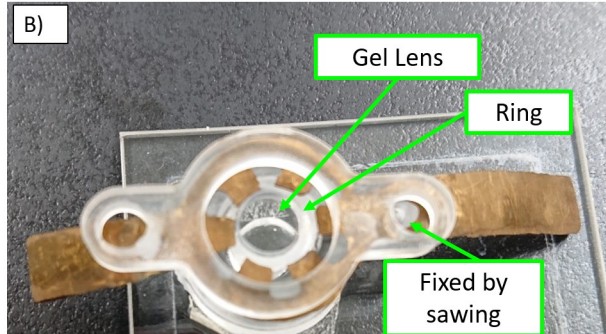

**Fig 6. Experimental prototype.** A) IPMC actuator. B) Prototype.

logic circuit, the eye on one side is in near-vision mode while the other is in far-vision mode. If a logic circuit is used, energy is only transferred when the electrical tool on both sides is attached. As such, the challenge of electrical energy transfer can be avoided.

## Experimental model

This manuscript describes a fundamental study. A prototype was made as shown in Fig 6.

**Problems encountered with swine lens.** In the previous study, we conducted experiments using the swine lenses as it was very difficult to make a lens soft and clear enough for use in optical experiments [15]. However, there are disadvantages with the use of swine lenses. They are not uniform as their diameters range from 10 mm to 13 mm, and the thicknesses are 9 mm to 12 mm, causing differences in experimental results. Secondly, swine lenses change their properties. Usually, the eye lens maintains its properties with the help of "hydatoid circulation". In cataract, this circulation is disabled. Inevitably, the swine lenses used in *in vitro* experiments are already devoid of "hydatoid circulation" with depleted supply of hydatoid. Experientially, after isolation from an eye, the lens shows cataract-like symptoms within 30 minutes. So, constant property maintenance of an artificial lens for experimental purposes is desired.

**Procedure for making the gel lens.** In this paper, we made a lens from silicone. As mentioned, it was difficult to make a lens soft and clear enough because a soft polymer is usually very sticky and weak. When the polymer is removed from the mold, it cracked easily and cannot be used for experiments. As an alternative plan, we made a lens with a methodology shown in Fig 7.

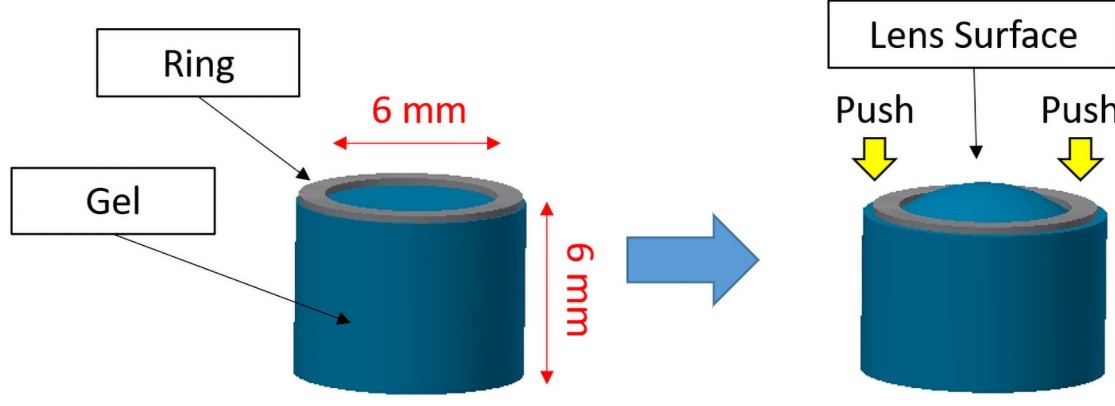

**Fig 7. Gel lens.**

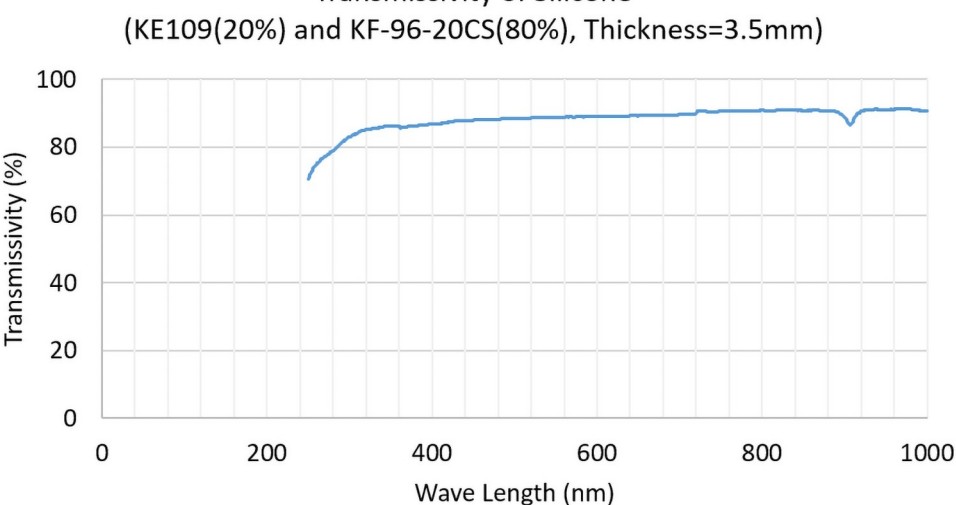

**Fig 8. Transmissivity of silicone.**

The silicone used was KE-109 (Shin-Etsu Silicone, Tokyo, Japan) which has two components. This silicone is made by mixing the two liquids under 100°C heat for longer than 1 hour. However, the usual type is too rigid to use as an accommodating IOL. In this experiment, KE-109 A liquid was mixed with B liquid and silicone oil KF-96-20CS (Shin-Etsu Silicone, Tokyo Japan) in the ratio 1:1:8 by weight. The mixture is then placed in an aluminum mold lined with soft Teflon film on the bottom and on the sides. We chose an aluminum mold, because aluminum has a high heat of conductivity. Teflon film was used as it has a higher detachability compared to hard materials such as glass. We heated the liquid mixture at 100°C for over 12 hours and obtained a form of columnar silicone. Then, we attached the hard ring and pressed it such that the shape of the inside of the ring was curved like a lens. This shape was not an actual spherical surface; however, it was superior to swine lenses for experimental purposes.

**Properties of the silicone.** We measured three properties of the silicone, i.e., transmissivity, refractive index, and Young's modulus.

Transmissivity is shown in Fig 8. We prepared the cuboid-shape silicone with a thickness of 3.5 mm. We placed it in air, applied 250 nm to 1000 nm wavelength light, and calculated the transmissivity of the silicone. In the visible-light area, transmissivity was higher than 85%, and considering that the Fresnel reflection is approximately 3%, it was actually almost 90%.

The refractive index is shown in Fig 9. We used an industrial microscope (USPM-RU-W, Olympus, Tokyo) and a ×10 objective lens (NA = 0.12) to measure the Fresnel reflection from a small surface area of the lens material. As the confocal microscope can measure the reflection of very small area, the measurement is not affected by the surface condition. We selected a flat and clean area to avoid including unwanted reflection components other than the Fresnel component. We repeated the measurements three times. Refractive indices were calculated from the measured reflection rates using the Fresnel reflection law. In the area of the optical wavelength, the refractive index was 1.40–1.41, which is a quite common value for the refractive index of silicone.

The Young's modulus of our silicone was 57.0 kPa. Considering that the Young's modulus of KE-109 silicone is 950 kPa, our silicone was more than 16 times softer than the original one. Actually, the rheologic properties are also important when considering such a very soft

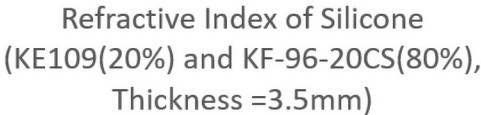

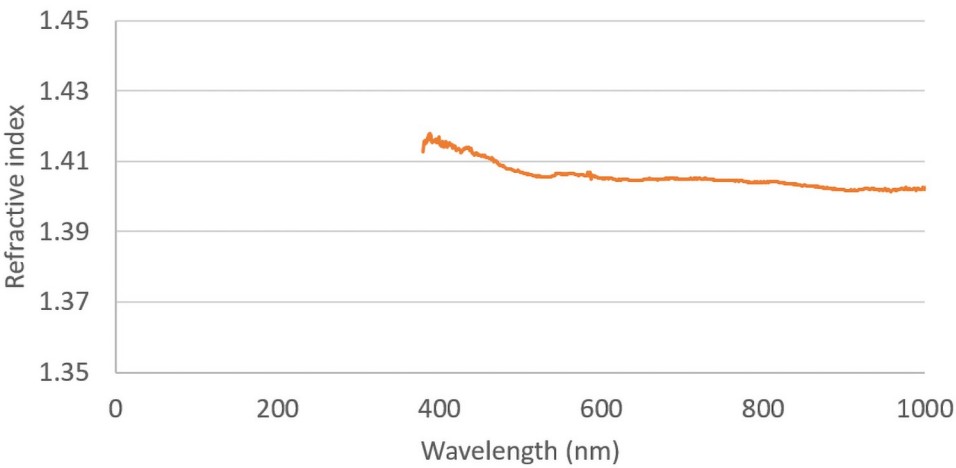

**Fig 9. Refractive index of silicone.**

material. However, our IOL system does not inflict large deformations to the lens, i.e., there is less than 0.1-mm deformation to a 6-mm thickness lens; therefore, we did not consider this.

**IPMC actuator.** The picture of an IPMC actuator is shown in Fig 6. It was made with Flemion (Asahi grass Inc., Tokyo, Japan), with a thickness of 0.15 mm and wing length of 2 mm. It is 40% of the previous IPMC actuator in length.

To make the IPMC actuator, Flemion membrane was roughened by sandblasting and was treated with gold-plated electrodes using an electrodeless plating process. For gold ions, we used gold phenanthroline liquid reduced with sodium sulfite.

**Experiments.** The experimental outline is shown in Fig 10. From the left, the camera had an accommodating IOL and diffuser panel. The camera was a compact Shack–Hartmann wavefront aberrometer (CSHWA) [16], which measured the power of the lens in diopters. The accommodating IOL was connected to an energy source, which was controlled by LabVIEW software (National Instruments, Austin, TX, USA). Square waves were applied at 0.1 Hz. The amplitude was ±1.25 to ±2.25 V.

## Results and discussion

The results are outlined in Fig 11 and Table 2. Maximum power obtained was +0.71 D. However, the actual minimum diopter cannot be determined due to noise.

### Maximum and minimum

Maximum power recorded was +0.71 D. The value was the same with 2.00 V and 2.25 V. This is demonstrated in Fig 12. The IPMC actuator did not touch the ring, and the power was not changed.

For minimum reading of power, reliable data could not be obtained. Voltages of 1.25 V and 1.50 V were not noisy, but 1.75 V resulted in a lot of noise. We think that this is because of the shape of the gel lens (Fig 13) that is different from the ideal spherical shape. When the actuator power is not too high, the surface is closer to a spherical surface. However, as the power

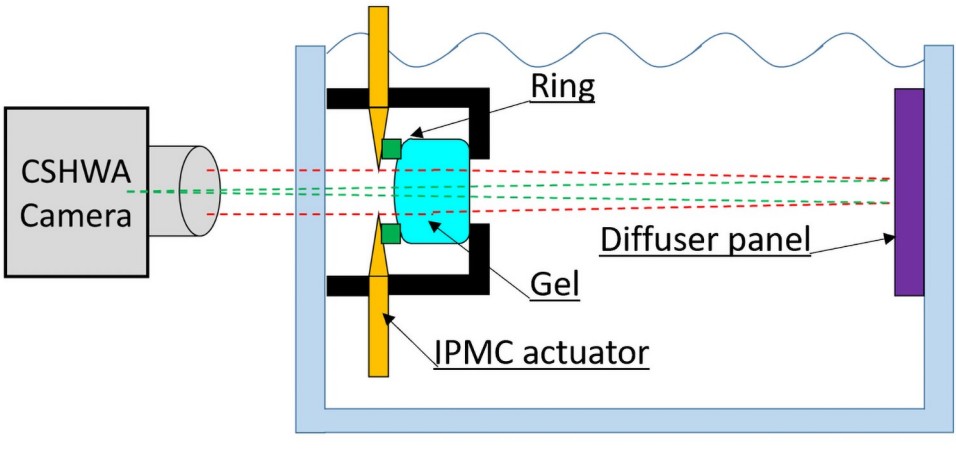

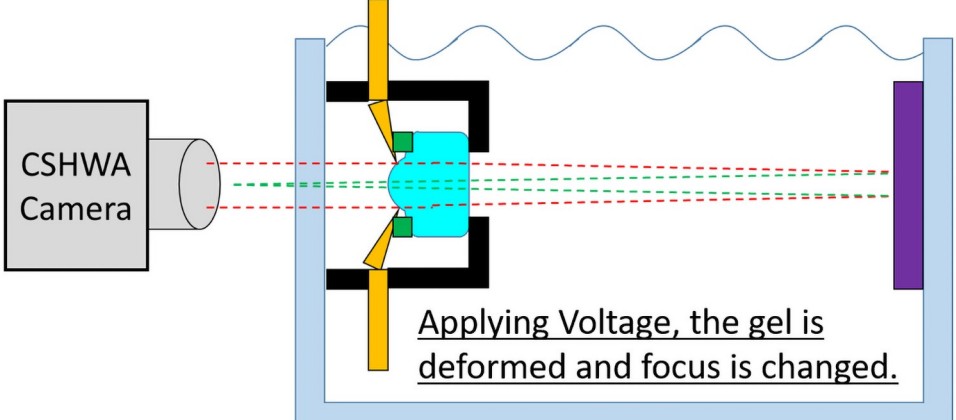

**Fig 10. Experimental situation.** CSHWA is compact Shake–Hartmann wavefront aberrometer, and IPMC actuator is ion polymer metal composite actuator.

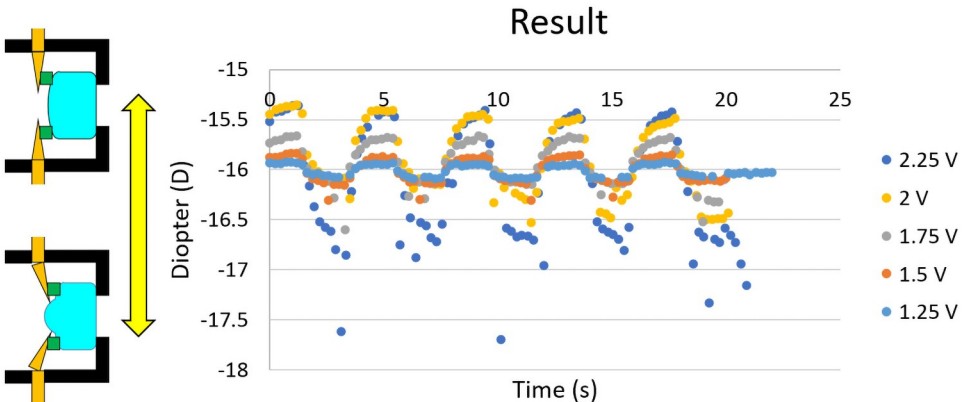

**Fig 11. Results of accommodating ranges in relation to time.**

**Table 2. Maximum and minimum power of the lens based on applied amplitude.**

| ↓Peak Voltage (V) | Max (D) | Min (D) |
|---|---|---|
| 1.25 | +0.15 | -0.02 |
| 1.50 | +0.23 | -0.23 |
| 1.75 | +0.41 | -0.53* |
| 2.00 | +0.71 | -0.46* |
| 2.25 | +0.71 | -1.62* |

*Data too noisy to assess.

increases, the radius of curvature of the center and near the ring becomes different. Thus, the calculated power becomes noisy. To minimize this, the shape of ring that attaches to the gel should be considered.

## Sixth order aberration

In Fig 14, we illustrate the sixth order aberration result. The sixth order terms of the Zernike polynomials are parameters that parameters that estimate the degree of sixth order aberration presence. In this study, the azimuth sixth order terms were very meaningful because they were found to be large in previous studies. This is because, previously, the lens was deformed by a six-winged actuator.

Additionally, by comparing the calculation results, we found that we could reduce Zernike 6. There was a 0.30-μm over 4-mm-radius circle area in our previous study, while it was 0.10 μm in our current study.

However, a second order aberration occurred which might have been caused by the inclination of the ring. The IPMC actuator was a soft actuator that could have been deformed by the

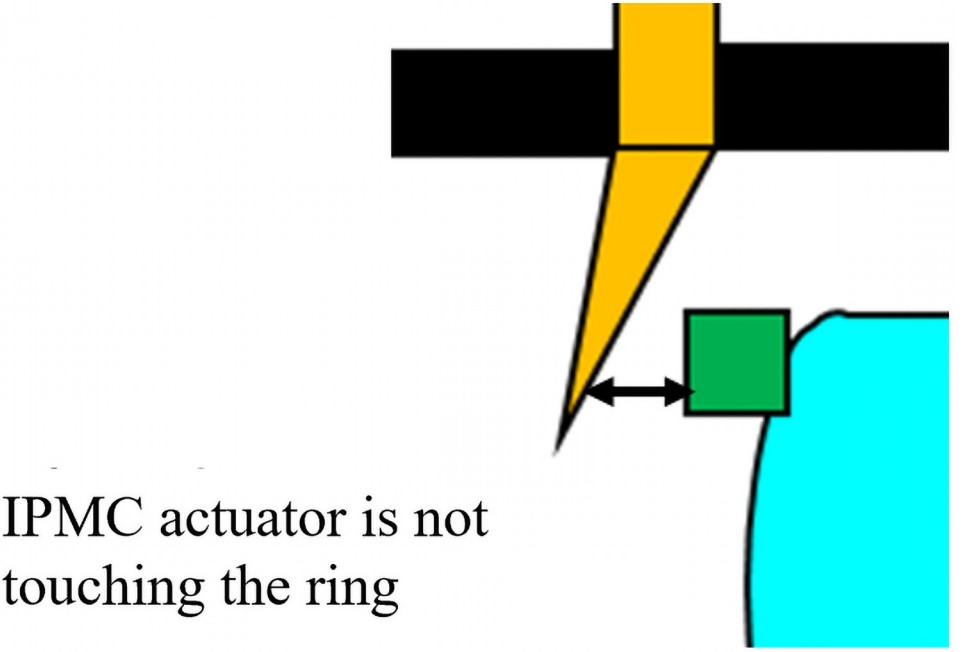

IPMC actuator is not touching the ring

**Fig 12. Upon application of more than 2.0 V, the IPMC does not touch the ring.** IPMC: Ion polymer metal composite.

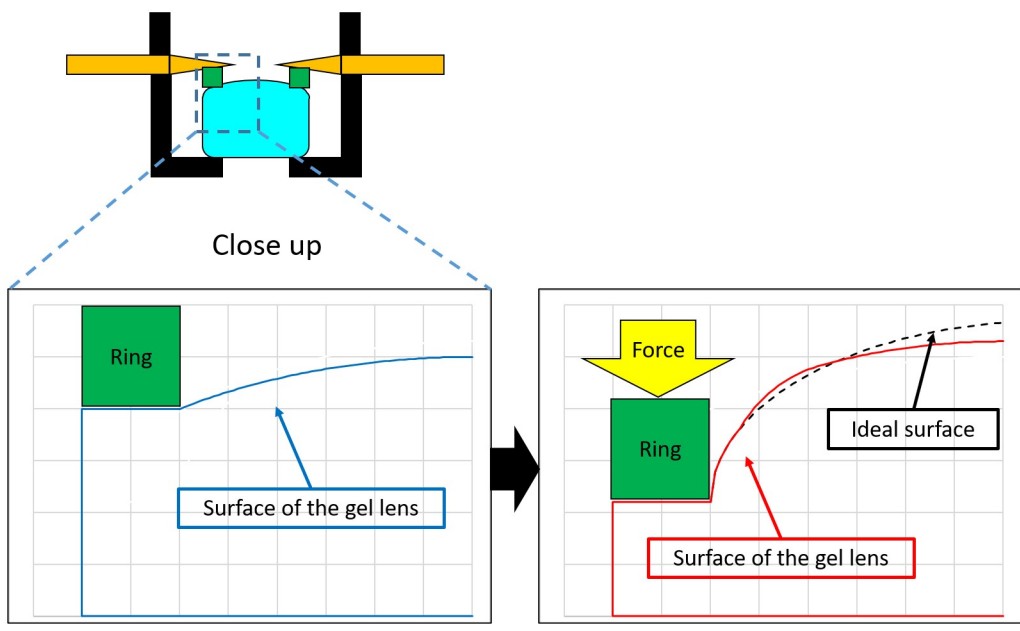

**Fig 13. Surface of the gel lens.**

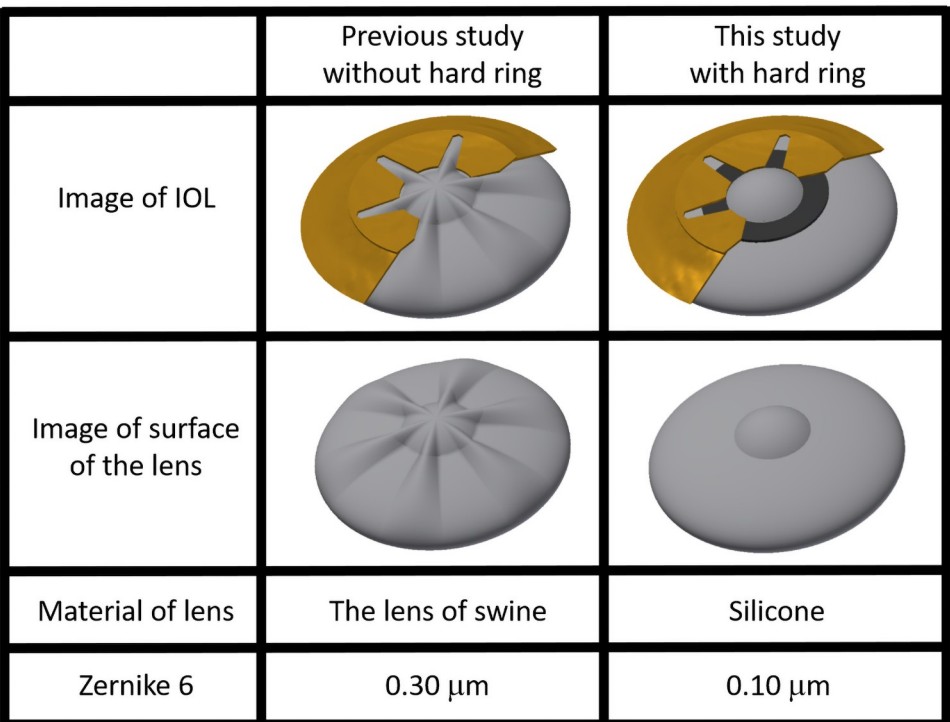

**Fig 14. Sixth order aberration.**

power of another side of the IPMC actuator. The power that pushes down the ring is also used to push up the other side of the ring. This not only produces a second order aberration, but also reduces the accommodating range.

## Smaller accommodating range

The other reason for the reduction in the accommodation range was that the size of the IPMC actuator was reduced. When the cation exchange capacity is constant, the power of the IPMC actuator is proportional to area and thickness of the actuator, reducing the power. Currently, we are using a gel lens; however, it is not soft enough. Several researchers (DeBoer et al. [17]) suggest using a deformable liquid balloon lens, which is softer than a gel lens. However, if a softer lens is used, there could be a relatively greater problem related to astigmatism. A careful deforming simulation will solve this problem.

## Biological compatibility

This was a pilot study, and, therefore, we did not select lens material with the objective of biological compatibility. However, to consider the surgical applications of this technology, we should discuss the coated layer. This coated layer requires high biological compatibility, has to be very soft and sufficiently thin to avoid preventing the movement of the IPMC actuator and lens, and transparent to allow light to pass. We watchfully await the results of the Fujie group's study [18] regarding this coated layer; they could create a membrane thinner than 1 μm, which would be biologically compatible. In the future, we should test an accommodating IOL system that would be fully covered by this film.

## Conclusion

We designed a new accommodating IOL system that has repeatability due to being fully artificial and, therefore, not requiring any biological material. Upon application of ±1.5 V, the power changed to ±0.23 D. This is smaller than our proposed accommodating range (±1.75 D), meaning that our artificial lens is not yet equal to its biological counterpart. However, we succeeded in eliminating the sixth-order wave aberration. This constitutes a considerable advancement on our previous design.

## Supporting information

**S1 File. This is supporting information file of the raw data of measurement of refractive index.**
(XLSX)

## Author Contributions

**Conceptualization:** Tetsuya Horiuchi.

**Data curation:** Tetsuya Horiuchi.

**Formal analysis:** Tetsuya Horiuchi.

**Funding acquisition:** Tetsuya Horiuchi.

**Supervision:** Tetsuya Horiuchi.

**Validation:** Tetsuya Horiuchi.

**Visualization:** Tetsuya Horiuchi.

**Writing – original draft:** Tetsuya Horiuchi.

**Writing – review & editing:** Tetsuya Horiuchi, Toshifumi Mihashi, Sujin Hoshi, Fumiki Okamoto, Tetsuro Oshika.

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
