## [Decision Letter · Decision Letter 0]

15 Feb 2021

PONE-D-20-40170

Artificial accommodating intraocular lens powered by an ion polymer-metal composite actuator

PLOS ONE

Dear Dr. Horiuchi,

Thank you for submitting your manuscript to PLOS ONE. After careful consideration, we feel that it has merit but does not fully meet PLOS ONE’s publication criteria as it currently stands. Therefore, we invite you to submit a revised version of the manuscript that addresses the points raised during the review process.

We look forward to receiving your revised manuscript.

Kind regards,

Tae-il Kim

Academic Editor

PLOS ONE

Reviewers' comments:

Reviewer's Responses to Questions

**Comments to the Author**

1. Is the manuscript technically sound, and do the data support the conclusions?

Reviewer #1: No

2. Has the statistical analysis been performed appropriately and rigorously? 

Reviewer #1: N/A

3. Have the authors made all data underlying the findings in their manuscript fully available?

Reviewer #1: No

4. Is the manuscript presented in an intelligible fashion and written in standard English?

Reviewer #1: No

5. Review Comments to the Author

Reviewer #1: The manuscript, “Artificial accommodating intraocular lens powered by an ion polymer-metal composite actuator” by Horiuchi et al., describes a novel design for an accommodating intraocular lens (IOL) for patients requiring cataract surgery. The design incorporates a deformable, artificial lens made of silicone with an actuator made of ion-polymer metal composite (IPMC). The device is connected to an energy source with a control system that ensures the actuator works only when the patient is focusing on a nearby object, while a hard ring placed between the actuator and lens reduces the sixth-order aberrations seen in earlier iterations of this design to second-order aberrations. Aberrometry measurements reveal the lens had a maximum dioptric power of +0.71 D at potential +2.00 V, and ±0.23 D at potential ±1.5 V. Although this falls short of the target accommodation range for the lens (±1.75 D), the authors believe the main benefits of this design are repeatability and cost effectiveness due to the fully artificial components, and elimination of high-order aberrations in the resulting image.

THE MAIN CLAIM AND SIGNIFICANCE OF THE MANUSCRIPT:

(1) This manuscript illustrates a novel design of IOL for an implement that combines innovations in both materials and biomedical engineering. The topic is interdisciplinary, and the result is highly interesting.

NOVELTY OF THE MANUSCRIPT:

(2) The lens design is based on the authors’ previous study [Horiuchi et al., Opt Express 24 (2016) 23280; REF 4 in the manuscript]. However, it is unclear and problematic why the authors did not cite their other work that is more closely related [Horiuchi et al., Smart Mater. Struct. 26 (2017) 045021].

(3) Based on their earlier works, using IPMC, system design are not new. Use of so-called ‘gel lens’ (instead of swine lens) seems to be a new idea from the authors’ perspective. This is a major material change and is novel. At the same time, information about this material must be provided properly – and this is an issue. Other claimed novelty includes the elimination of sixth-order aberration by using the material (although this term was not properly defined in the manuscript.)

(4) Generally speaking, this manuscript lacks critical information in many important fields so that the manuscript leaves many unanswered questions that hinder its ability to argue its significance and novelty. Writing itself have many issues to be considered for academic publication.

DATA ANALYSIS SUPPORTING THE CLAIMS

(5) Related with point (3), the authors must provide critical material information related with their so-called ‘gel lens’. What is the transparency vs wavelength? (UV-vis spectra must have been provided at the least) What is the mechanical property of the silicone KE-109 A mixed with KF-96-20-CS that was mixed at 1:1:8 ratio? (Rheological measurement information must have been provided) Did it have a ‘jacket’ layer to contain any unreacted silicone from oozing out? What was the refractive index of the ‘gel lens’, as a function of wavelength? Was it, at least, consistent? Whereas the ‘gel lens’ is the key for the novelty of this manuscript, there are no information about these in this manuscript.

(6) Lines 63-65 sum up the main problem according to the authors: “The conventional accommodating IOL can be used to control the focus of the lens with the patient’s ciliary muscle; however, a significant problem exists. The accommodating ranges vary widely from patient to patient. This phenomenon cannot be resolved by these methods while the patients’ ciliary muscles are used for accommodating”. Two questions I have are:

a) There is no source provided for this. How large is the variance in accommodating range among patients? i.e. how many patients struggle with using accommodating IOLs due to lack of ciliary muscle function?

b) Etiology of presbyopia is actually disputed [DeBoer et al., IEEE Transactions on Biomed. Eng., 64 (6), 1129-1135 (2016); REF 14 in the manuscript] and one theory suggests that presbyopia occurs primarily due to hardening of the lens itself. Conceivably, in some cases patients retain most ciliary muscle function at the time of cataract surgery. Is this type of actuator-based IOL still beneficial or even viable for patients of this nature?

(7) Although it arguably led to the most significant improvement in results (elimination of sixth-order aberrations) the design features and properties of the “hard ring” are never fully examined and discussed. Is the composition of this material significant? For example, “Then, we attached the hard ring and pressed it such that the shape of the inside of the ring was curved like a lens” (lines 143-144) is quite difficult to visualize. Closer examination of the part geometry around the ring (and the ring itself) would be helpful (maybe in Figure 3, 6, or 7), especially since changes to the shape of the ring are suggested as a potential improvement to prevent noise.

(8) It is suggested that a potential improvement is switching to a liquid balloon lens. However, earlier in the article it was mentioned that liquid crystal lenses were unsafe to use at voltages >1.5 V due to electrical leakage. This suggests that the experimental conditions used in this manuscript (voltages up to 2.25 V) would not be feasible if the switch to a liquid lens was made.

(9) What is the precedent for embedding a stable and non-degrading power source, complete with added logic circuit, in the eye sockets? It seems that such a design would be prone to eventual failure with so much movement from the eyeball occurring in the area. Is it also possible that this will prohibitively increase the complexity and complications of the cataract surgery?

(10) The IPMC that the authors used: What is the time required to adjust the focal length? Is the IPMC reaction time comparable to what our eyes can do?

WRITING ISSUES

(10) In addition to the lack of the necessary details, the literary value of the manuscript itself must improve before being considered for publication. Introduction and Conclusion must flesh out better. The results and discussion parts needs reorganization after including the necessary details provided above.

REFERENCING ISSUES

(11) This manuscript does not give fair treatment for other group’s recent works. For example, a recent review paper [Mylona and Tsinopoulos, Pharmaceauticals 13 (2020) 448 and 80 references therein] includes recent advances in the field of IOL. There have been many exciting studies on silicone-based lens with artificial muscles and material analysis, for example [Maffi et al., Adv Funct Mater 25 (2015) 1614; Carpi et al., Adv Funct Mater 21 (2011) 4152].

MINOR ERRORS

(12) Small typo (also present in previous paper by this author), “NuLans” should be NuLens [Alio et al., J. Cataract and Refractive Surgery, 35 (10), 1671-1678 (2008)].

(13) Refs [5] and [6] citation sequence?

(14) “hydatoid circulation” � explanation needed for general readers

(15) “Usually, silicone is made by mixing the two liquids under 100oC … too hard to use as an accommodating IOL.” � I believe that the authors refer to Sylgard 184. There are many silicones – just going to local hardware store can assure that there are many soft silicones curing at room temperature.

6. PLOS authors have the option to publish the peer review history of their article (what does this mean?). If published, this will include your full peer review and any attached files.

Reviewer #1: **Yes: **Hyun-Joong Chung

---

## [Author Response · Author response to Decision Letter 0]

30 Mar 2021

Thank you for reviewer and editor, and my answer is written in attached file.

Black letters sentences are your comments.

 Purple letters sentences are my answers.

 Red letters sentences are sentence which is written in my paper.

---

## [Decision Letter · Decision Letter 1]

20 Apr 2021

PONE-D-20-40170R1

Artificial accommodating intraocular lens powered by an ion polymer-metal composite actuator

PLOS ONE

Dear Dr. Horiuchi,

Thank you for submitting your manuscript to PLOS ONE. After careful consideration, we feel that it has merit to be published in Plos One. However, reviewer pointed out one minor issue. Therefore, we invite you to submit a revised version of the manuscript that addresses the points raised during the review process.

Reviewer comments were addressed.  More supporting data could have been revealed (for example, where are the evidences for the transmittance 'almost 90%'?)

We look forward to receiving your revised manuscript.

Kind regards,

Tae-il Kim

Academic Editor

PLOS ONE

Journal Requirements:

Reviewers' comments:

Reviewer's Responses to Questions

**Comments to the Author**

1. If the authors have adequately addressed your comments raised in a previous round of review and you feel that this manuscript is now acceptable for publication, you may indicate that here to bypass the “Comments to the Author” section, enter your conflict of interest statement in the “Confidential to Editor” section, and submit your "Accept" recommendation.

Reviewer #1: All comments have been addressed

2. Is the manuscript technically sound, and do the data support the conclusions?

Reviewer #1: Partly

3. Has the statistical analysis been performed appropriately and rigorously? 

Reviewer #1: N/A

4. Have the authors made all data underlying the findings in their manuscript fully available?

Reviewer #1: No

5. Is the manuscript presented in an intelligible fashion and written in standard English?

Reviewer #1: Yes

6. Review Comments to the Author

Reviewer #1: Reviewer comments were addressed. More supporting data could have been revealed (for example, where are the evidences for the transmittance 'almost 90%'?)

7. PLOS authors have the option to publish the peer review history of their article (what does this mean?). If published, this will include your full peer review and any attached files.

Reviewer #1: No

---

## [Author Response · Author response to Decision Letter 1]

21 Apr 2021

Thank you your comment and I attached supporting data.

This is raw data for measuring refractive index, which is including data of Fresnel reflection.

---

## [Editor Report · Decision Letter 2]

27 May 2021

Artificial accommodating intraocular lens powered by an ion polymer-metal composite actuator

PONE-D-20-40170R2

Dear Dr. Horiuchi,

We’re pleased to inform you that your manuscript has been judged scientifically suitable for publication and will be formally accepted for publication once it meets all outstanding technical requirements.

Kind regards,

Tae-il Kim

Academic Editor

PLOS ONE

---

## [Editor Report · Acceptance letter]

14 Jun 2021

PONE-D-20-40170R2 

Artificial accommodating intraocular lens powered by an ion polymer-metal composite actuator 

Dear Dr. Horiuchi:

I'm pleased to inform you that your manuscript has been deemed suitable for publication in PLOS ONE. Congratulations! Your manuscript is now with our production department. 

Kind regards, 

on behalf of

Dr. Tae-il Kim 

Academic Editor

PLOS ONE